# Dysfunctional Beliefs and Cognitive Performance across Symptom Dimensions in Childhood and Adolescent OCD

**DOI:** 10.3390/jcm12010219

**Published:** 2022-12-28

**Authors:** Federica Piras, Nerisa Banaj, Valentina Ciullo, Fabrizio Piras, Giuseppe Ducci, Francesco Demaria, Stefano Vicari, Gianfranco Spalletta

**Affiliations:** 1Neuropsychiatry Laboratory, Department of Clinical and Behavioral Neurology, IRCCS Santa Lucia Foundation, Via Ardeatina 306, 00179 Rome, Italy; 2Mental Health Department, ASL Roma 1, Piazza Santa Maria della Pietà 5–Pad. 26, 00193 Rome, Italy; 3Child and Adolescent Neuropsychiatry Unit, Department of Neuroscience, Bambino Gesù Children’s Hospital, IRCCS, Piazza S. Onofrio 4, 00165 Rome, Italy; 4Department of Life Sciences and Public Health, Università Cattolica del Sacro Cuore, Largo Francesco Vito 1, 00168 Rome, Italy; 5Menninger Department of Psychiatry and Behavioral Sciences, Baylor College of Medicine, 1977 Butler Blvd., Houston, TX 77030, USA

**Keywords:** Obsessive–Compulsive Disorder, obsessive beliefs, OCD symptom dimensions, pediatric OCD, cognitive performance

## Abstract

Although etiological and maintenance cognitive factors have proved effective in predicting the disease course in youths with OCD, their contribution to symptom severity and specific OCD dimensions has been scarcely examined. In a cohort of children and adolescents with OCD (N = 41; mean age = 14; age range = 10–18 yrs.), we investigated whether certain dysfunctional beliefs and cognitive traits could predict symptom severity, and whether they were differentially associated with specific symptom dimensions. We found that self-oriented and socially prescribed perfectionism and intolerance to uncertainty were associated with higher obsession severity, which was not uniquely related to any neuropsychological variable. Greater severity of obsessions and compulsions about harm due to aggression/injury/violence/natural disasters was predicted by excessive concerns with the expectations of other people. Severity in this dimension was additionally predicted by decreasing accuracy in performing a problem-solving, non-verbal reasoning task, which was also a significant predictor of severity of obsessions about symmetry and compulsions to count or order/arrange. Apart from corroborating both the belief-based and neuropsychological models of OCD, our findings substantiate for the first time the specificity of certain dysfunctional beliefs and cognitive traits in two definite symptom dimensions in youth. This bears important clinical implications for developing treatment strategies to deal with unique dysfunctional core beliefs, and possibly for preventing illness chronicity.

## 1. Introduction

Obsessive–Compulsive Disorder (OCD) is a clinical heterogenous illness characterized by multiple and temporally stable symptom dimensions [1]. Growing evidence suggests that symptom dimensions of OCD have unique patterns of heritability [2,3] and neurobiological correlates [4,5] with possibly, a differential course of illness [6] and treatment response [7,8]. As a matter of fact, in a recent investigation, pediatric OCD patients with symmetry/hoarding symptoms showed less a favorable response to pharmacotherapy [9], while a meta-analysis targeting mixed age patients indicated that the hoarding subtype had a poorer response to cognitive-behavioral therapy compared with the non-hoarding subtype [10]. This evidence reinforces the notion that the dimensional approach may be clinically valuable in orienting more personalized treatment strategies considering the possibility that specific symptom dimensions are subtended by definite early behavioral precursors.

With the intent to provide insights on the pathogenetic process leading to the full manifestation of OCD symptoms, different theoretical models have been proposed. To date, the most well-articulated and accounted model is the cognitive-behavioral model [11,12] (namely, the belief-based cognitive theory) positing that the misinterpretation of intrusive thoughts plays a pivotal role in triggering the development of clinical obsessions, which determine the engagement in compulsive behavior so as to resist, block, or neutralize these obsessive thoughts. More specifically, intrusive thoughts, which are also experienced by 80–90% of the general population [13], are catastrophically misinterpreted in OCD such as personally significant or otherwise highly threatening. This leads to elevated distress and efforts to reduce the perceived threat and consequent distress via compulsive rituals and avoidance. Misinterpretations of intrusive thoughts are supposed to arise from a set of dysfunctional beliefs (The Obsessive Compulsive Cognitions Working Group [14]) including the tendency to overestimate threat and personal responsibility, the excessive concern about the importance of controlling one’s thoughts, and an exaggerated need for perfectionism and certainty. Previous studies [15,16,17] indicate that OCD symptom dimensions may be subtended by specific obsessive beliefs. Specifically, the contamination/cleaning dimension would be associated with overestimation of threat [16,18] and inflated responsibility [18,19], the forbidden thoughts dimension (i.e., aggression, sexual, religious, and somatic obsessions and checking compulsions) with over-importance and need to control thoughts [18,19], and the symmetry dimension (i.e., symmetry obsessions and repeating, ordering, and counting compulsions) with the need for perfectionism and certainty [17,18]. Among others, the perfectionism belief is the most pervasive, accounting for different symptom dimensions [18,20], and also being present in non-affected siblings of OCD patients and in non-clinical samples exhibiting obsessive compulsive personality traits. Cognitive models of OCD propose that obsessive beliefs may develop in childhood and are central to the etiology and maintenance of the disorder [11]. Specifically, dysfunctional beliefs, and particularly the inability to tolerate mistakes or imperfection and to accept uncertainty or ambiguity, are related to variation in the disorder severity [21] and could predict forthcoming OCD symptom severity in a non-clinical sample [22]. Despite this evidence, thus far, scanty experimental research has investigated the contribution of these dysfunctional beliefs to symptom severity and specific OCD dimensions in both adult and pediatric samples. Determining whether maladaptive perfectionism is associated with specific pediatric OCD symptom dimensions and symptom severity is a crucial issue for identifying early predictors of OCD.

In parallel, in the effort to validate neuropsychological models of the disorder [23,24,25] in the past two decades, a large number of neuropsychological studies have reported cognitive deficits as one of the main characteristics of patients with OCD [26,27]. However, in contrast to the extensive literature on cognition in adult OCD, the characterization of the neuropsychological profile in the pediatric population is inconclusive. Although some studies have reported deficits in specific cognitive domains [28,29,30,31], contrarily to the adult OCD presentation, a disorder-specific neuropsychological profile in pediatric OCD is yet to be determined [32]. However, recent evidence [33] demonstrated that deficiencies in specific subdomains of cognitive flexibility and inhibitory control, namely, initial concept formation and proactive control, may be valid candidate neurocognitive endophenotypes of OCD being observable in pediatric probands, their unaffected siblings, and parents. Multiple hypotheses explain the divergence between pediatric and adult studies, and it has been suggested that neuropsychological deficits subtended by developmental disruption to brain maturation processes [24,34] may become evident only as the disorder progresses [29]. Conversely, neural maturation often brings improvement or remission of some OCD symptoms, with diminished comorbid conditions (i.e., tics) and improved executive function [35] contributing to further differentiating the pediatric and adult cognitive profiles. Indeed, neuroimaging studies support the developmental component of OCD, and different patterns of cortical and subcortical abnormalities have been identified in pediatric and adult patients [36,37], thus explaining the reported difference in neuropsychological functioning. Such clinical, familial, and translational biomarker correlates seen in early onset OCD would support the notion of a developmental subtype of the disorder [35]. Nonetheless, inconsistence in results could also be related to several factors, such as the small sample sizes, strict exclusion criteria, patients’ comorbidities, and symptoms heterogeneity, which limit generalizability of findings. Particularly, an optimal methodology to reconcile such disparate evidence would be the dimensional subtyping approach, as emerging evidence [38,39,40] suggested that unique neuropsychological deficits may characterize specific symptom clusters. While such a hypothesis has been scarcely investigated in pediatric samples, conflicting findings emerge from studies on adult OCD patients since the aggressive and contamination dimensions were related to poorer performance on tasks of cognitive inhibition [40,41], and the symmetry dimension was associated with impaired performance on memory tasks [40], while other studies found different results [42,43].

All the above considered, and based on the assumption that tracing (and treating) early predictors of symptom severity in the pediatric community may reduce the potential for negative effects over time, here, we examined the clinical profile of a fairly large cohort of children and adolescents with OCD. We aimed at investigating whether certain dysfunctional beliefs and cognitive traits could predict symptom severity, and whether they were differentially associated with specific symptom dimensions in youth OCD. Understanding these associations in childhood will foster better outcomes through targeted and individualized treatments, and consequently prevent chronicity and persistence of OCD symptomatology into adulthood.

## 2. Materials and Methods

### 2.1. Participants

To determine the sufficient sample size for measuring potential correlations between multiple variables within one sample case, a power analysis (*t*-test for a point biserial correlational model two tailed, alpha = 0.05 and power = 0.9) was conducted using G*Power [44] and a clinically significant effect size of 0.5. Based on the aforementioned assumptions, the minimum desired sample size was 34. Here, we included 41 subjects with OCD.

Patients with a DSM-5 [45] diagnosis of OCD were enrolled at the Children’s Research Hospital and IRCCS “Bambino Gesù” and the Mental Health Department—ASL Roma 1, and assessed at the Neuropsychiatry Laboratory of IRCCS Santa Lucia Foundation in Rome (Italy). Diagnosis of OCD was made by two child and adolescent neuropsychiatrists and confirmed using the Structured Clinical Interview for DSM-5 Disorders- Clinician Version (SCID-5-CV) [46] by a research psychiatrist. Inclusion criteria were: (1) age between 10 and 18 years; (2) OCD as primary diagnosis; and (3) Italian language native speaker. Exclusion criteria were: (1) intellectual disability [45] as measured by standardized cognitive assessment (the Wechsler Intelligence Scale for Children, Fourth edition and the Wechsler Adult Intelligence Scale—Fourth Edition) [47,48] (i.e., IQ < 70); (2) traumatic brain injury with loss of consciousness; (3) past or present major medical illnesses, such as diabetes, asthma, hematological/oncological disorders, clinically significant and unstable active gastrointestinal, renal, hepatic, endocrine or cardiovascular system disease, hypothyroidism; (4) presence of any brain abnormality and micro-vascular lesion apparent on conventional T1-weighted and FLAIR-scans [49]; (5) history of psychoactive substance dependence or abuse, schizophrenia-spectrum disorders and bipolar disorders as investigated by the SCID-5-CV [46]; and (6) significant changes in current medication within the last 6 weeks.

### 2.2. Experimental Procedure

All subjects underwent a comprehensive evidence-based psychopathological and neuropsychological assessment expressly conceived for OCD. The psychopathological assessment included: (a) the Children’s Yale–Brown Obsessive–Compulsive Disorder Scale (CY-BOCS) [50] and the Dimensional Yale–Brown Obsessive–Compulsive Disorder Scale (DY-BOCS) [51] for assessing severity of OCD symptoms (Obsession sub-score, Compulsion sub-score and Total score) and for providing a detailed description of obsessions and compulsions that are divided into seven different OC dimensions (i.e., Contamination/Cleaning, Hoarding/Collecting, Symmetry/Order, Violence/Disaster, Sex/Religious, Somatic and Miscellaneous dimensions); (b) the Child–Adolescent Perfectionism Scale (CAPS) [52], a 22-items measure based on the multidimensional conceptualization of perfectionism for evaluating Self-Oriented Perfectionism (SOP) and Socially Prescribed Perfectionism (SPP); (c) the Intolerance of Uncertainty Scale for Children (IUSC) [53], a 27-item questionnaire for measuring children’s tendency to negatively react in the context of uncertain situations and events; (d) the Penn-State Worry Questionnaire for Children (PSWQ-C) [54] to assess the tendency of youth to engage in excessive, generalized, and uncontrollable worry; (e) the Children Depression Inventory (CDI) [55], which rates the severity of symptoms related to depression or dysthymia in children and adolescents; and (f) the Children’s Global Assessment Scale (CGAS) [56], a measure of general functioning. The neuropsychological test battery included: (a) the Raven’s Colored Progressive Matrices (RCPM) [57] for measuring abstract reasoning and non-verbal intelligence; (b) the Stroop Color and Word Test (SCWT) [58] to assess basic attentional abilities and processing speed and the ability to inhibit the cognitive interference that occurs when the processing of a specific stimulus feature impedes the simultaneous processing of a second stimulus attribute [59]; (c) the Go/No Go task [60] aimed at determining the ability to inhibit a response deemed as inappropriate; (d) the Rey–Osterrieth Complex Figure test (ROCF) [61] for visual perception and long term visual memory; (e) the Delayed Item Recognition task (DIR) [62] to measure the maintenance component of visual non-verbal working memory; and (f) the Tower of London (BACS version) [63] to assess non-verbal reasoning and problem solving capacities.

### 2.3. Data Analysis

Significance of correlations of psychopathological and neuropsychological variables was measured using Fisher’s r to z transformation. A statistical model corrected for multiple comparisons according to the False Discovery Rate (FDR) procedure was used to control for type I (false positive) error. Specifically, each individual *p*’s Benjamini–Hochberg critical value was calculated using the formula (*i*/*m*)*Q*, where: *i* = the individual *p*-value’s rank, *m* = total number of tests (N = 276), and *Q* = the false discovery rate (*p* < 0.05).

In order to identify the specific patterns of dysfunctional beliefs and neuropsychological processes characterizing OCD symptom dimensions, separate forward stepwise multiple regression analyses (*F* > 4 to enter) were further performed including only symptom dimensions, measures of dysfunctional beliefs, and cognitive performance which resulted as significantly correlated in the previous analyses. Indeed, the forward stepwise procedure starts with no variables in the model and it tries out the variables one by one, including them if they are statistically significant, and thus identifying the best set of predictors that gives the biggest improvement to the model. The Variance Inflation Factor (VIF) [64] was calculated for each model to investigate multicollinearity between variables.

Correlational analyses were performed on StatView statistical software. Values that remained significant after FDR correction <0.05 were identified using an online calculator (https://tools.carbocation.com/FDR, accessed on 4 May 2022), while SPSS (IBM SPSS Statistics for Macintosh, Version 27.0) was used for regression models and multicollinearity testing.

## 3. Results

### 3.1. Demographic and Clinical Characteristics

The characteristics of the sample are shown in Table 1. Only total scores from Symmetry/Order, Contamination/Cleaning, and Violence/Disaster dimensions were included in the analyses, as few patients in our sample endorsed symptoms pertaining to the other dimensions. Such a distribution is largely consistent with previous studies evaluating the frequency of most common symptom dimensions in children and adolescents patients with OCD [65].

The majority of patients (N = 34) had other psychiatric disorders in addition to OCD, specifically: 34% generalized anxiety disorder, 24% mood disorders, 22% attenuated psychotic syndrome, 20% tic disorders, 15% attention deficit hyperactivity disorder, and 5% eating disorders. Most patients were under stable pharmacological treatment (71%), while 12% of patients were drug naive.

### 3.2. OCD Symptoms, Dysfunctional Beliefs and General Functioning

Table 2 reports the complete correlation matrix. Once corrected for multiple comparisons according to FDR, the three scores of the CY-BOCS were positively correlated with the Symmetry/Order dimension (total score: r = 0.6; *p* = 0.007; obsession sub-score: r = 0.5; *p* = 0.003; compulsive sub-score: r = 0.6; *p* = 0.003). In parallel, positive correlations were found between the CY-BOCS obsession sub-score and measures of perfectionism, specifically the CAPS total score (r = 0.5; *p* = 0.007) and its sub-scores measuring the perfectionistic self-presentation (i.e., SOP, requirement of the self to be perfect) (r = 0.5; *p* = 0.02) and the interpersonal expression of perfectionism (i.e., SPP, perception that others require perfection of oneself) (r = 0.4; *p* = 0.03). Moreover, the CY-BOCS obsession sub-score was positively correlated with the intolerance of uncertainty (IUSC) score (r = 0.4; *p* = 0.03). Lastly, the CY-BOCS showed negative correlations with a measure of general functioning (CGAS) (total score: r = −0.7; *p* = 0.002; obsession sub-score: r = −0.7; *p* = 0.005; compulsive sub-score: r = −0.6; *p* = 0.002).

### 3.3. OCD Symptom Dimensions, Dysfunctional Beliefs and Cognitive Profile

Specific significant correlations were found for two OCD symptom dimensions, measures of dysfunctional beliefs and cognitive performance. In particular, the Violence/Disaster dimension was significantly correlated with the SP perfectionism measure (r = 0.5; *p* = 0.003), with the IUSC-intolerance of uncertainty score (r = 0.4; *p* = 0.031) and with accuracy in the Tower of London task (r = −0.4; *p* = 0.034). Ancillary analyses demonstrated that the severity distribution of aggression/injury/violence/natural disasters symptoms did not vary as a function of gender (*p* > 0.05), while age and illness duration were not significant predictors of symptom severity in this dimension (*p* > 0.05). Conversely, the Symmetry/Order dimension showed negative significant correlations with the Tower of London accuracy (r = −0.4; *p* = 0.044) and CGAS (r = −0.5; *p* = 0.012) scores. See Table 3 for the complete results corrected for multiple comparisons and Table 4 for a comprehensive view of the main correlational findings.

Results from the first stepwise regression, including the Violence/Disaster dimension as the dependent variable, and SPP sub-score, IUSC, and Tower of London scores as independent variables, showed that two predictors could significantly explain 38% of observed variance in symptom severity (F_2,38_ = 13.32; *p* < 0.0001; R^2^ = 0.412; adjusted R^2^ = 0.381). Specifically, the SPP sub-score (β = 0.5; *p* < 0.0001) and the Tower of London accuracy score (β = −0.4; *p* = 0.009) were significant predictors of severity in this specific cluster of symptoms, while the IUSC score was excluded from the analysis (β = 0.2; *p* = 0.202). The observed VIF was <2 for each variable indicating no apparent multicollinearity between measures. Conversely, results from the regression analysis including the Symmetry/Order dimension as the dependent variable and the Tower of London accuracy score as the independent variable showed that the latter was a significant predictor of symptomatology severity (F_1,38_ = 8.103; *p* = 0.007; R^2^ = 0.172; adjusted R^2^ = 0.151) and could explain 15% of the total variance observed in this OCD dimension (β = −0.4; *p* = 0.007). Figure 1 depicts the significant relationships between severity scores in specific OCD dimensions and cognitive/dysfunctional belief measures.

## 4. Discussion

The bimodal incidence distribution of OCD, with a peak in childhood [66] and a second one during early adulthood [67], and its consequent detrimental impact on functioning in maturity (if not properly treated), fostered intense research on early predictors of symptom severity in the community of children and adolescents [22,68]. Factors acknowledged as crucial for the etiology and maintenance of OCD in cognitive [69,70] and neuropsychological [23,24,25] models of the disorder have proved effective in prospectively predicting the disease course in children and early adolescents [21,22]. Specifically, dysfunctional beliefs, and particularly the inability to tolerate mistakes or imperfection and to accept uncertainty or ambiguity, are related to variations in the disorder severity [21] and could predict forthcoming OCD symptom severity in a non-clinical sample [22]. In parallel, premorbid impairment in visuospatial abilities and some forms of executive dysfunction during adolescence were associated with adult OCD diagnosis in the most well-known longitudinal study on multidisciplinary health and development [24]. Particularly, executive deficits in concept formation and proactive control, both of which may be valid candidate endophenotypes of the disorder, have been associated with intolerance to ambiguity in OCD, as performance improves when elements of ambiguity regarding task demands are resolved [33]. Thus, need for perfection in behavior to overcome feelings of uncertainty [71], and neuropsychological deficits subtended by developmental disruption to brain maturation processes [24,34], in addition to being possibly intertwined, have been identified as early precursors and catalysts of OCD.

### 4.1. Relationship between OCD Symptom Dimensions and Dysfunctional Beliefs

In a cohort of children and adolescents with OCD we indeed found that self-oriented and socially prescribed perfectionism and intolerance to uncertainty were related to obsessions severity, thus confirming the belief-based model of childhood OCD. However, no neuropsychological correlate was found for severity of compulsive and obsessive symptoms. Further, by adopting the quantitative multidimensional approach to OCD symptoms [72] in childhood and early adolescence, we investigated whether the unique patterns of dysfunctional beliefs [15,16,18,20,73] and cognitive traits [38,40] characterizing symptom dimensions of OCD in adults were maintained in a pediatric sample. We found that greater severity in the symptomatology dimension characterized by thematically related obsessions and compulsions about harm due to aggression/injury/violence/natural disasters [51,72] was predicted by excessive concerns with the expectations of other people (i.e., socially prescribed perfectionism). Decreasing accuracy in mentally performing a problem-solving, non-verbal reasoning task [63] (i.e., the Tower of London test) also predicted increasing obsessions and compulsions severity in this dimension. Likewise, greater severity of obsessions about symmetry and more frequent/time consuming and disabling compulsions to count or order/arrange [51,72] were predicted by decreasing accuracy in the same problem-solving task.

Although it’s fairly well established that the structure of OCD symptoms seen in adults is preserved in pediatric samples [51,74,75], very little is known about the association between particular obsessive beliefs and specific themes of OCD symptoms in children and early adolescents [65]. In adult samples, OCD symptom dimensions are significantly predicted by obsessive beliefs above and beyond general negative affectivity, although substantial unexplained variance advocates for the involvement of additional factors [18]. Moreover, inconsistencies in the literature also deriving from heterogeneity in the adopted measures and constructs for assessing either symptom dimensions [20] or the cognitive phenomena involved in OCD suggest the possibility of cross-cultural variations [20,65]. Indeed, the observation that aggression and harm-related obsessions and compulsions are linked to beliefs about the importance of, and need to control thoughts (a construct that was not tested in the present sample) is not universal [14]. Perfectionism can actually predict the aggression and harm-related dimension [20], particularly in studies involving OCD patients with checking compulsions as results of obsessions related to harm due to aggression/violence/disasters [69,76]. Equally, although OCD symptom dimensions seem to have unique beliefs domains [18], cultural factors may intervene in this association, such that obsession and compulsion severity in specific dimensions is not predicted by any dysfunctional belief [20].

The scanty literature exploring the incidence of dysfunctional beliefs in children and adolescents with OCD, stratified according to symptomatologic dimensions [65], revealed that more than 2/3rd of subjects endorsed some obsessive beliefs (i.e., avoidance, pathological doubting, overvalued sense of responsibility, pervasive slowness, and indecisiveness, none of which was considered in the present investigation), but did not explore the relationship between beliefs and symptom dimensions. In a recent pediatric OCD investigation [77], cognitive beliefs appeared to be relevant for obsessions about harm/responsibility and checking compulsions. Although a large significant correlation was observed between perfectionism, intolerance to uncertainty, and a dimension (doubting/checking) corresponding to harm/responsibility obsessions, perfectionism/uncertainty beliefs were not associated to the same dimension when all belief domains were combined in a single multivariate model [77]. However, different symptom dimensions and dysfunctional belief classifications were employed as derived from self-report measures of OCD symptoms (i.e., the Obsessive–Compulsive Inventory [78]) and cognitive beliefs (i.e., Obsessive Beliefs Questionnaire [79]), potentially explaining the dissimilar findings. Thus, the present evidence, demonstrating that excessive concerns with the expectations of other people explain a significant portion of variance in the severity of aggression and harm-related symptoms, constitutes the first quantitative substantiation of such a pattern of dysfunctional beliefs in this specific dimension [20] in childhood and early adolescence. In another prospective study on youths [22], besides being an early predictor of OCD symptoms severity, perfectionism predicted doubting/checking, washing, ordering, and neutralizing compulsions. However, in this study, subjects were classified into six theme-based dimensions based on the form, rather than the function fulfilled by symptoms [18]. In contrast, in the here adopted measure for evaluating specific OCD dimensions [52], symptoms were assessed according to their purpose; for example, ordering rituals performed to reduce fear of causing harm were ascribed to the aggression and harm-related dimension, not to the symmetry one. Thus, the trans-dimensional assessment of compulsions, and the different method in classifying symptoms into theme-based dimensions may justify the different finding in the present investigation. Additionally, the fact that, here, a clinical sample (as opposed to children and early adolescents recruited from the community [22]) was assessed might explain the difference in results concerning the significant effect of perfectionism on the severity of aggression and harm-related symptoms. Indeed, it can be argued that the observed relationship between dysfunctional beliefs and overall severity of obsessive-compulsive symptoms might hold true only in those with such a decline in functioning as to seek treatment for symptoms relief. As a matter of fact, since only subjects with a full manifestation of symptoms were included, the same effect might not be observable in individuals experiencing subclinical OC symptomatology, although sparse evidence suggests that, at least in OC subclinical adults, some obsessive-compulsive beliefs are related to symptoms [80]. Despite the above-mentioned limitations, this is the first study investigating the dysfunctional beliefs profile of the different symptomatology dimensions in youth with OCD using more than one conceptual framework to measure perfectionism. Actually, the CAPS here adopted was developed to investigate motivational and cognitive referents that focus primarily on perfectionistic standards that come from the self or others (i.e., Self-Oriented Perfectionism and Socially Prescribed Perfectionism), and there are few scale items that assess self-evaluations in terms of the ability or inability to attain these standards [52]. Thus, it is fundamental to further characterize in future studies the impact of the maladaptive externally motivated need for perfection in originating specific themes of symptoms.

### 4.2. OCD Symptom Dimensions and Cognitive Profile

Regarding the correlation between neuropsychological performance and symptom dimensions in OCD, although both attentional and executive functioning deficits are the main features of the disorder [81], whether specific dysfunctions are correlated with certain symptom dimensions remains unclear. Indeed, previous investigations have largely ignored the possibility that cognitive disturbances may vary across OCD clinical dimensions, or the potentiality of an intertwined relationship between dysfunctional beliefs, cognitive impairments, and dimensional symptoms [82]. Moreover, as far as we know, few studies [83,84] have explored this issue in clinical samples of children and adolescents with OCD. Our finding of a significant influence of executive performance on symptom severity in the aggression/harm-related and symmetry dimensions may corroborate theories arguing that specific neuropsychological deficits in childhood (especially in problem solving and non-verbal reasoning [24]) contribute to OCD clinical manifestation. However, this might hold true only for specific clusters of symptoms. In adult samples, in the aggression/harm-related [40] and symmetry dimensions [38,85] an inverse relationship was observed between symptom severity and executive performance. Concurrently, in treatment-seeking youth diagnosed with OCD [83], patients in the symmetry dimension had a greater magnitude of cognitive impairment, with specific executive and processing speed deficits, possibly explained by the supposed attentional bias toward symmetrical and/or ordered stimuli that may draw their attention away from task performance. However, in another study, the proportion of patients with aggression/harm-related and symmetry symptoms within the small group showing neuropsychological impairment was not different from the proportion of patients in the unimpaired group [84]. Moreover, considering such scanty and inconclusive evidence, we must assume that the present finding points toward potential trans-dimensional variations in neuropsychological performance that need to be systematically studied in larger samples also exploring the neurobiological correlates of poor cognitive performance and clinical presentation.

Before concluding, some limitations of the present study should be considered. First, no strict direction of causal relationship can be drawn between OCD symptoms, dysfunctional beliefs, and neuropsychological processes using a cross-sectional sample and regression analyses. However, although based on a multiple regression approach, the present evidence comes from a clinical sample of OCD patients selected according to very stringent exclusion criteria, with no intellectual developmental disorders or neurological illnesses. Thus, the unique patterns of dysfunctional beliefs and neuropsychological profile characterizing the aggression/harm-related and symmetry dimensions may be considered observable traits (phenotypes) that can be discerned in clinical practice. Additionally, although the investigated sample is relatively large, post hoc power analyses for the significant influence of obsessive beliefs and cognitive functioning on the emergence of specific clusters of obsessive symptoms demonstrated that a true impact was detectable in almost the entirety of the included subjects (99% and 78%). Such results strengthen the reported evidence suggesting that it might be generalized to larger populations of treatment-seeking OCD youths.

A second point is that the majority of the included patients had comorbid psychiatric disorders in addition to OCD, thus potentially weakening the specificity of our findings. Indeed, maladaptive perfectionism has been described as a transdiagnostic vulnerability factor for several disorders including, but not limited to, depression, anxiety, and eating disorders [86,87]. The fact that patients typically present at least one or multiple co-occurring disorders [88] makes the issue of comorbidity an intrinsic problem in this field of psychiatric research. Nevertheless, in the subgroup of OCD patients with clinically significant comorbid generalized anxiety disorder (i.e., GAD, mostly coexisting with other psychiatric disorders), anxiety symptoms severity did not contribute to explaining the severity of aggression/injury/violence/natural disasters symptoms. Additionally, no significant association (FDR corrected) was observed between severity of depressive symptoms (as measured using the Children Depression Inventory) and the variables of interest, suggesting that self-reported depression (within the sub-clinical range) was not related to OCD symptom severity, nor to maladaptive beliefs and cognitive functioning. Likewise, it might be argued that other demographic/clinical variables (such as gender, age, and illness duration) may have impacted the OCD symptom profiles, thus affecting the here observed relationship between dysfunctional beliefs, cognitive abilities, and symptom dimensions. However, the severity distribution of aggression/injury/violence/natural disasters symptoms did not vary as a function of gender, while age and illness duration were not significant predictors of symptom severity in this dimension. Therefore, the selectivity of our finding with a precise link between perfectionism and a symptom dimension specifically related to the disorder suggests that this relationship may be distinctive of early onset OCD. Lastly, it could be argued that we did not explore the influence of the whole pattern of dysfunctional beliefs in our sample of OCD patients, focusing only on the need for perfectionism and certainty. However, a recent longitudinal study [22] found perfectionism/certainty beliefs to be early predictors of OCD symptoms in youth, while beliefs regarding threat overestimation, inflated responsibility, and over-importance/control of thoughts were less consistently observed, thus raising questions concerning the role and specificity of these constructs in OCD symptoms development for the young age group.

## 5. Conclusions

Notwithstanding these limitations, our study is the first substantiation suggesting the contribution of two dysfunctional beliefs and cognitive ability to the overall and content-specific severity of OCD symptoms in childhood and early adolescence. Indeed, while maladaptive perfectionism and intolerance to uncertainty were associated with higher obsession severity, socially prescribed perfectionism was related to severity in a specific cluster of symptoms (aggressive obsessions and compulsions), which was also linked to decreased accuracy in a problem-solving task. The latter also had a significant influence on the severity of symmetry obsessions and related compulsions. The present finding bears important clinical implications as specific rather than generic cognitive-behavioral treatment strategies targeting the unique core beliefs underlying individual symptom dimensions may be developed. In parallel, addressing neuropsychological weaknesses in childhood with interventions targeting dimension-specific deficits may divert the illness course; ascertainment and early intervention in affected youth may abbreviate functional impairments associated with the untreated illness [35] and might prevent chronicity/persistence into adulthood.

## Figures and Tables

**Figure 1 jcm-12-00219-f001:**
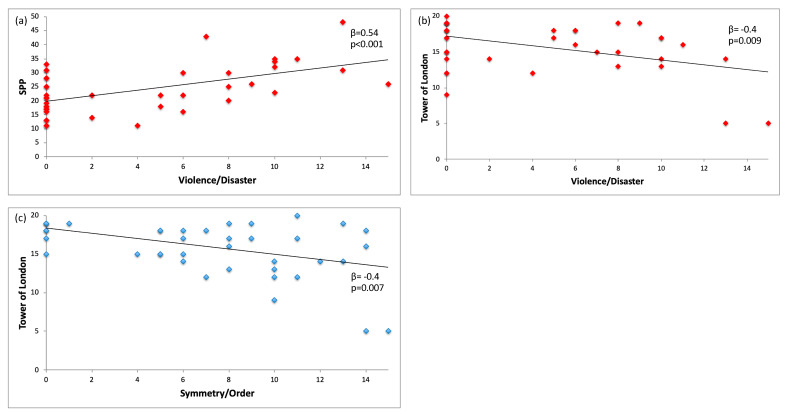
Relationships between OCD dimensions, cognitive and dysfunctional belief measures: (**a**) Violence/Disaster dimension and SPP scores; (**b**) Violence/Disaster dimension and Tower of London total scores; (**c**) Symmetry/Order dimension and Tower of London total scores. Legend: SPP: Socially Prescribed Perfectionism.

**Table 1 jcm-12-00219-t001:** Sociodemographic and clinical characteristics of 41 early onset OCD patients.

	OCD (N = 41) Mean (SD)
Age	14 (2.2)
Education	7.8 (2)
Gender (males)	N = 30 (73%)
Illness duration (years)	4.2 (3.1)
CY-BOCS tot	23.1 (8.6)
CY-BOCS obsessive	11.7 (4.3)
CY-BOCS compulsive	11.4 (4.5)
Symmetry/Order	7.3 (4.5)
Contamination/Cleaning	4.8 (5)
Violence/Disaster	4.1 (4.7)
Hoarding/Collecting	2.6 (3.4)
Miscellaneous	2.6 (3.5)
Somatic	2.5 (3.9)
Sex/Religious	2.1 (3.7)
CDI tot	13.9 (7.2)
CAPS tot	58.7 (17.3)
SOP	34.8 (10.8)
SPP	23.9 (8.6)
IUSC	67.2 (26.4)
PSWQ-C	34.4 (8.7)
CGAS	49.4 (9.2)

Notes: SD: standard deviation; CY-BOCS: Child Yale–Brown Obsessive–Compulsive Scale; CDI: Child Depression Inventory; CAPS: Child–Adolescent Perfectionism Scale; SOP: Self-Oriented Perfectionism; SPP: Socially Prescribed Perfectionism; IUSC: Intolerance of Uncertainty Scale for Children; PSWQ-C: Penn-State Worry Questionnaire for Children; CGAS: Children’s Global Assessment Scale.

**Table 2 jcm-12-00219-t002:** Correlation Matrix for clinical and cognitive variables for the whole sample.

	CY-BOCS Tot	CY-BOCS Obsessive	CY-BOCS Compulsive	Contam. Cleaning	Symmetry Order	Violence/ Disaster	IUSC	CAPS Tot	SOP	SPP	PSWQ-C	CDI	CGAS	SCWT Word	SCWT Colour	SCWT Interf.	Go/No Go Hit	Go/No Go Mean RTs	RCPM	ROCF Copy	ROCF Recall	Tower of London	DIR Hit	DIR Mean RTs
CY-BOCS tot	1.0	1.0	1.0	0.4	0.6	0.3	0.4	0.4	0.3	0.4	0.3	0.4	−0.7	0.0	0.1	0.1	0.0	−0.1	−0.2	0.0	0.2	−0.2	0.0	−0.1
CY-BOCS obsessive	1.0	1.0	0.9	0.4	0.5	0.4	0.4	0.5	0.5	0.4	0.3	0.4	−0.7	0.1	0.1	0.0	0.0	−0.1	−0.1	0.0	0.2	−0.2	0.0	−0.1
CY-BOCS compulsive	1.0	0.9	1.0	0.4	0.6	0.2	0.3	0.3	0.2	0.2	0.3	0.4	−0.6	−0.1	0.1	0.1	−0.1	−0.1	−0.2	0.1	0.2	−0.2	0.0	0.0
Contam./ Cleaning	0.4	0.4	0.4	1.0	0.0	0.1	0.0	0.2	0.1	0.3	0.1	−0.2	−0.3	−0.1	0.0	−0.2	−0.2	−0.1	0.0	−0.2	0.1	0.0	0.1	−0.1
Symmetry Order	0.6	0.5	0.6	0.0	1.0	0.3	0.3	0.4	0.4	0.3	0.4	0.2	−0.5	−0.1	0.0	0.1	0.2	0.2	−0.2	−0.1	−0.1	−0.4	0.2	−0.1
Violence/ Disaster	0.3	0.4	0.2	0.1	0.3	1.0	0.4	0.4	0.3	0.5	0.4	0.2	−0.3	0.2	0.4	0.2	0.1	0.2	−0.2	−0.2	−0.1	−0.4	0.0	−0.3
IUSC	0.4	0.4	0.3	0.0	0.3	0.4	1.0	0.7	0.6	0.6	0.7	0.5	−0.3	0.1	−0.1	−0.2	0.3	0.1	0.0	0.1	0.0	−0.1	0.2	−0.2
CAPS tot	0.4	0.5	0.3	0.2	0.4	0.4	0.7	1.0	0.9	0.9	0.5	0.2	−0.4	0.2	0.1	0.0	0.2	0.0	0.0	0.1	0.1	−0.1	0.4	−0.4
SOP	0.3	0.5	0.2	0.1	0.4	0.3	0.6	0.9	1.0	0.6	0.5	0.2	−0.3	0.3	0.1	0.0	0.2	0.1	0.0	0.1	0.1	0.0	0.3	−0.3
SPP	0.4	0.4	0.2	0.3	0.3	0.5	0.6	0.9	0.6	1.0	0.4	0.1	−0.3	0.1	0.0	0.1	0.2	−0.1	0.0	0.0	0.2	−0.2	0.3	−0.4
PSWQ-C	0.3	0.3	0.3	0.1	0.4	0.4	0.7	0.5	0.5	0.4	1.0	0.5	−0.3	0.2	0.2	−0.2	0.1	0.4	−0.2	0.0	−0.1	−0.1	0.3	0.0
CDI	0.4	0.4	0.4	−0.2	0.2	0.2	0.5	0.2	0.2	0.1	0.5	1.0	−0.2	0.0	−0.1	−0.1	0.2	0.1	−0.1	0.0	−0.1	−0.1	0.1	−0.1
CGAS	−0.7	−0.7	−0.6	−0.3	−0.5	−0.3	−0.3	−0.4	−0.3	−0.3	−0.3	−0.2	1.0	−0.1	−0.1	−0.1	0.1	−0.2	0.2	0.1	0.1	0.3	0.1	0.2
SCWT word	0.0	0.1	−0.1	−0.1	−0.1	0.2	0.1	0.2	0.3	0.1	0.2	0.0	−0.1	1.0	0.7	0.6	-0.4	0.2	-0.3	−0.1	−0.1	0.0	-0.4	0.3
SCWT colour	0.1	0.1	0.1	0.0	0.0	0.4	−0.1	0.1	0.1	0.0	0.2	−0.1	−0.1	0.7	1.0	0.6	-0.3	0.2	-0.5	-0.2	-0.2	-0.2	-0.4	0.3
SCWT interf.	0.1	0.0	0.1	-0.2	0.1	0.2	−0.2	0.0	0.0	0.1	−0.2	−0.1	−0.1	0.6	0.6	1.0	−0.1	0.2	−0.4	−0.3	−0.2	−0.4	−0.4	0.2
Go/No Go hit	0.0	0.0	−0.1	−0.2	0.2	0.1	0.3	0.2	0.2	0.2	0.1	0.2	0.1	−0.4	−0.3	−0.1	1.0	0.0	0.2	0.2	0.1	0.0	0.4	−0.4
Go/No Go mean RTs	−0.1	−0.1	−0.1	−0.1	0.2	0.2	0.1	0.0	0.1	−0.1	0.4	0.1	−0.2	0.2	0.2	0.2	0.0	1.0	0.0	−0.1	−0.3	−0.2	0.0	0.1
RCPM	−0.2	−0.1	−0.2	0.0	−0.2	−0.2	0.0	0.0	0.0	0.0	−0.2	−0.1	0.2	−0.3	−0.5	−0.4	0.2	0.0	1.0	0.3	0.2	0.5	0.4	−0.2
ROCF copy	0.0	0.0	0.1	−0.2	−0.1	−0.2	0.1	0.1	0.1	0.0	0.0	0.0	0.1	−0.1	−0.2	−0.3	0.2	−0.1	0.3	1.0	0.7	0.2	0.3	−0.2
ROCF recall	0.2	0.2	0.2	0.1	−0.1	−0.1	0.0	0.1	0.1	0.2	−0.1	−0.1	0.1	−0.1	−0.2	−0.2	0.1	−0.3	0.2	0.7	1.0	0.2	0.3	−0.4
Tower of London	−0.2	−0.2	−0.2	0.0	−0.4	−0.4	−0.1	−0.1	0.0	−0.2	−0.1	−0.1	0.3	0.0	−0.2	−0.4	0.0	−0.2	0.5	0.2	0.2	1.0	0.1	0.0
DIR hit	0.0	0.0	0.0	0.1	0.2	0.0	0.2	0.4	0.3	0.3	0.3	0.1	0.1	−0.4	−0.4	−0.4	0.4	0.0	0.4	0.3	0.3	0.1	1.0	−0.2
DIR mean RTs	−0.1	−0.1	0.0	−0.1	−0.1	−0.3	−0.2	−0.4	−0.3	−0.4	0.0	−0.1	0.2	0.3	0.3	0.2	−0.4	0.1	−0.2	−0.2	−0.4	0.0	−0.2	1.0

Notes: CY-BOCS: Child Yale–Brown Obsessive–Compulsive Scale; IUSC: Intolerance of Uncertainty Scale for Children; CAPS: Child–Adolescent Perfectionism Scale; SOP: Self-Oriented Perfectionism; SPP: Socially Prescribed Perfectionism; PSWQ-C: Penn State Worry Questionnaire for Children; CDI: Child Depression Inventory; CGAS: Children’s Global Assessment Scale; SCWT: Stroop Color and Word Test; RCPM: Raven’s Colored Progressive Matrices; ROCF: Rey–Osterrieth Complex Figure copy and delayed recall; DIR: Delayed Item Recognition task.

**Table 3 jcm-12-00219-t003:** False Discovery Rate results.

	Original *p* Value	Critical Value	Benjamini-Hochberg Adjusted *p* Value
CY-BOCS tot, DY-BOCS Symmetry/Order	0.0001	0.0007	0.0069
CY-BOCS obsessive, DY-BOCS Symmetry/Order	0.0002	0.0038	0.0026
CY-BOCS compulsive, DY-BOCS Symmetry/Order	0.0001	0.0016	0.0031
CY-BOCS obsessive, CAPS tot	0.0006	0.0042	0.0072
CY-BOCS obsessive, SOP	0.0023	0.0054	0.0212
CY-BOCS obsessive, SPP	0.0032	0.0056	0.0285
CY-BOCS obsessive, IUSC	0.0038	0.0062	0.0308
CY-BOCS obsessive, CGAS	0.0001	0.0009	0.0055
CY-BOCS compulsive, CGAS	0.0001	0.0024	0.0021
DY-BOCS Violence/Disaster, SPP	0.0002	0.0036	0.0028
DY-BOCS Violence/Disaster, IUSC	0.0037	0.0060	0.0309
DY-BOCS Violence/Disaster, Tower of London	0.0046	0.0067	0.0343
DY-BOCS Symmetry/Order, CGAS	0.001	0.0043	0.0115
DY-BOCS Symmetry/Order, Tower of London	0.0065	0.0074	0.0438

Notes: CY-BOCS: Child Yale–Brown Obsessive–Compulsive Scale; DY-BOCS: Dimensional Yale–Brown Obsessive–Compulsive Scale; CAPS: Child–Adolescent Perfectionism Scale; SOP: Self-Oriented Perfectionism; SPP: Socially Prescribed Perfectionism; IUSC: Intolerance of Uncertainty Scale for Children; CGAS: Children’s Global Assessment Scale.

**Table 4 jcm-12-00219-t004:** Summary of correlational analysis results FDR corrected.

OCD Symptoms	Dysfunctional Beliefs	Cognitive Traits
Higher obsessions severity	Increased self-oriented perfectionism Increased socially prescribed perfectionism Increased intolerance to uncertainty	
Greater severity of obsessions and compulsions about harm due to aggression/injury/violence/natural disasters	Increased socially prescribed perfectionism Increased intolerance to uncertainty	Decreased accuracy in a problem-solving, non-verbal reasoning task
Greater severity of obsessions about symmetry, and compulsions to count or order/arrange		

## Data Availability

All authors take full responsibility for the data, the analyses and interpretation, and the conduct of the research. They gave consent for full data access and the right to publish any and all data. They also confirm that neither the manuscript nor any parts of its content are currently under consideration or published in another journal.

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
