# Peer review of "Dysfunctional Beliefs and Cognitive Performance across Symptom Dimensions in Childhood and Adolescent OCD"

_jcm, 2022, doi:10.3390/jcm12010219_

Round 1
Reviewer 1 Report
Thank you for the opportunity to revise this manuscript. The authors assessed central beliefs (perfectionism, intolerance of uncertainty) central neurocognitive measures (attention, inhibition, problem solving), and clinical measures (general functioning, OCD severity, and the severity of several OCD symptom dimensions in 41 children and adolescents diagnosed with OCD. Main results indicate that perfectionism and intolerance had significant positive associations with severity of obsessions. Specific perfectionism subscale was associated with violence\disaster subscale. Accuracy in a problem solving task (Tower of London) exhibited a negative correlation with severity of symmetry symptoms and counting\arranging symptoms. The authors suggest that their results corroborate existing OCD models and bare possible clinical implications. I think this manuscript may replicate and extend existing findings once some issues are addressed. Please see below for my suggestions.
Major comments
1. I applaud the authors for including an a priori power calculation. This is important in any study, but especially in pediatric OCD studies that are often underpowered. The authors should explain their rationale for excepting an effect size of 1 and which effect size are they referring to. The authors should also state the number of variables they include in the a-priori or post-hoc power calculation. The number of predictors would impact the power calculation.
2. The experimental design is cross-sectional, but the authors suggest that their results support or validate OCD models. This claim should be revised or substantially unpacked.
3. The discussion cites multiple sources. I believe the authors should curate these results and structure the discussion paragraphs so that the claims and their supporting evidence are consistent. It would help if the authors used more consistent wording. For example, is there a difference between “symptom domains” and “subtypes”?
4. Proof reading is advised.
Minor comments
Abstract
1. How do the results “confirm both the belief-based and neuropsychological models of OCD…”?
Introduction
2. P.1, first sentence – I suggest the authors revise the “temporally stable” statement or cite a longitudinal study supporting this claim.
3. P.1. line 42-47 – The cited hoarding findings contributed to the creation of hoarding disorder diagnosis in the DSM-5 and ICD-11. It is uncertain whether these individuals would have been characterized with OCD today. In a way CBT for OCD is personalized as the exposures are tailored according to the symptoms. Do the authors believe that there should be different therapeutic interventions for different symptom domains? If so, please state additional evidence supporting this claim.
4. P.2. – When describing the CBT model, I suggest replacing “anxiety” with “distress” in line with recent developments in the field.
5. P.2, line 91 – Please add references
Method
6. P.3, line 138 – Please state the standardized cognitive assessment name. Are the authors referring to “QI” or “IQ”.
7. P.3, line 142 – Did all patients undergo a FLAIR scan?
8. P.3, line 153-155 – Please add details on the dimensional YBOCS so it is clear whether each dimension captures both obsessions and compulsions.
9. P.4, line 167 – For the Stroop task – please consider using a “interference” minus “word” measure instead of the three measures described.
10. P.3, line 182 – What were the 276 tests?
11. P.3, line 183-5 – What variables were the authors trying to predict and why? In which order were the predictors added to the regression?
12. P.3, line 185 – What “F” are the authors referring to and how did they choose the value 4 as a crucial value?
Results
13. The revised power analysis should include the number of predictors the authors test in this section.
14. Please provide more details on the stepwise regression.
15. Were there any correlations between the neuropsychological measures and OCD severity?
Discussion
16. P.12, line 325 – the authors mention “belief-based model”. This is the first time they use this term. Please explain it.
17. P.12, lines 334-336 – I suggest the authors revise this sentence for readability.
18. P.13, lines 390-392 – What are the different conceptual perfectionism frameworks used in this study?
19. P.12, line 340-P.13, line 394 – I wonder if the authors can split this paragraph to two paragraphs focusing on symptom dimensions and on perfectionism?
20. P.13, line 403 – Please rephrase “predictive effect”
21. P.13, line 403 – The authors suggest this is an “executive performance” problem, but there were no associations between the target variable and different executive measures, such as the Stroop task. I think it would be useful to explain what the accuracy measure in the “Tower of London” task represents and discuss differences between this measure and the other executive measures used in this sample.
22. P.14, lines 442-445 – I might have missed it, but does this information appear in the “methods” and “results” sections? If not, please add it.

Author Response
Please, see attachment

Reviewer 2 Report
This is a very interesting original research work about dysfunctional beliefs and cognitive performance across symptom dimensions in pediatric OCD. Congratulations to the authors for their clarity of exposure, scientific rigor and capacity to summarize findings. I only have minor comments to address:
1- Although methods and results are clearly detailed, presented and summarized, it is still difficult for a non-expert reader to grasp the key highlights of the study. I’d suggest to create a table or a figure integrating information on 1. OCD symptoms (obsessions or compulsions or both), 2. dysfunctional beliefs (perfectionism, expectations of, uncertainty intolerance, etc.), and 3. cognitive traits (decreased performance on problem solving, etc). Below an example, although maybe a figure could be more illustrating:
|
OCD |
Dysfunctional beliefs |
Cognitive traits |
|
Higher obsessions severity |
· self-oriented · socially prescribed perfectionism · intolerance to uncertainty |
|
|
Greater severity of obsessions and compulsions about harm due to aggression/injury/violence/natural disasters |
excessive concerns with the expectations of other people
|
decreasing accuracy in performing a problem-solving, non-verbal reasoning task |
|
Severity for obsessions about symmetry, and compulsions to count or order/arrange |
|
2- for the power analysis, you considered a clinically significant effect size of 1, which may be quite a lot of expected effect size and thus reduce the sample size required. Could you please elaborate on how did you decide this number (arbitrarily, based on the literature…), and accordingly state it on limitations if it is one.
3- when performing multiple comparisons, you state that corrections were performed:
· “A statistical model corrected for multiple comparisons according to the False Discovery Rate (FDR) procedure was used to control for type I (false positive) error.”
· “Correlational analysis and multiple comparison correction were performed on Stat-View statistical software, while SPSS (IBM SPSS Statistics for Macintosh, Version 27.0) was used for regression models and multicollinearity testing. “
However, it is still not clear to me which method was used specifically. Could you please elaborate on that part?
4- You use repeatedly the concept of “prediction”, instead of “association”, but you state in limitations that “no strict direction of causal relationship can be drawn between OCD symptoms, dysfunctional beliefs and neuropsychological processes using a cross-sectional sample and regression analyses”.
Considering that, wouldn’t it be more accurate to talk about associations instead of predictions? I’d recommend to tone a little bit down the strength of some discussion statements of the results considering the design and limitations, although I believe you discuss it appropriately.
5- Regarding presence of psychiatric comorbidities in the majority of patients: “ (N = 34) had other psychiatric disorders in addition to OCD, specifically: 34 % Generalised Anxiety Disorder, 24 % mood disorders, 22 % Attenuated Psychotic Syndrome, 20 % Tic Disorders, 15 % Attention Deficit Hyperactivity Disorder and 5 % Eating Disorders.”.
You already discuss the fact that this point weakens the findings of the study, and you point out that anxiety symptoms in the OCD group were not severe. Did you assess in any way the presence and severity of symptoms of anxiety, depression, psychosis, etc in a transdiagnostic way? Or excluded patients with a primary diagnosis not being OCD? Could you please elaborate this point apart from anxiety and OCD as you have already done? Thanks!
6- The conclusion seems too general, could you add a sentence or two referring to the main results?

Author Response
Please, see the attachment

Round 2
Reviewer 2 Report
The authors have addressed major points properly.